# Effects of Three Different Doses of Inter-Alpha Inhibitor Proteins on Severe Hypoxia–Ischemia-Related Brain Injury in Neonatal Rats

**DOI:** 10.3390/ijms232113473

**Published:** 2022-11-03

**Authors:** Liam M. Koehn, Kevin Nguyen, Xiaodi Chen, Andre Santoso, Richard Tucker, Yow-Pin Lim, Barbara S. Stonestreet

**Affiliations:** 1Women & Infants Hospital of Rhode Island, Alpert Medical School of Brown University, Providence, RI 02905, USA; 2ProThera Biologics, Inc., Providence, RI 02905, USA

**Keywords:** hypoxia–ischemia, neonate, brain, encephalopathy, drug, pediatric, efficacy, behavior, asphyxia, infarct volume

## Abstract

Hypoxia–ischemia (HI)-related brain injury is an important cause of morbidity and long-standing disability in newborns. We have previously shown that human plasma-derived inter-alpha inhibitor proteins (hIAIPs) attenuate HI-related brain injury in neonatal rats. The optimal dose of hIAIPs for their neuroprotective effects and improvement in behavioral outcomes remains to be determined. We examined the efficacy of 30, 60, or 90 mg/kg of hIAIPs administered to neonatal rats after exposure to HI for 2 h. Postnatal day 7 (P7) Wistar rats were exposed to either sham-surgery or unilateral HI (right carotid artery ligation, 2 h of 8% O_2_) brain injury. A placebo, 30, 60, or 90 mg/kg of hIAIPs were injected intraperitoneally at 0, 24 and 48 h after HI (*n* = 9–10/sex). We carried out the following behavioral analyses: P8 (righting reflex), P9 (negative geotaxis) and P10 (open-field task). Rats were humanely killed on P10 and their brains were stained with cresyl violet. Male extension/contraction responses and female righting reflex times were higher in the HI placebo groups than the sham groups. Female open-field exploration was lower in the HI placebo group than the sham group. hIAIPs attenuated these behavioral deficits. However, the magnitude of the responses did not vary by hIAIP dose. hIAIPs reduced male brain infarct volumes in a manner that correlated with improved behavioral outcomes. Increasing the hIAIP dose from 30 to 90 mg/kg did not further accentuate the hIAIP-related decreases in infarct volumes. We conclude that larger doses of hIAIPs did not provide additional benefits over the 30 mg/kg dose for behavior tasks or reductions in infarct volumes in neonatal rats after exposure to severe HI.

## 1. Introduction

Hypoxia–ischemia (HI) can damage the neonatal brain by limiting access of essential oxygen and nutrients to the brain and stimulating destructive systemic and neuroinflammatory cascades. Brain injury is known to progress over days to weeks after the initial onset of HI in neonates [1,2], suggesting that there could be a relatively wide therapeutic window of opportunity for the application of neuroprotective treatments. Hypothermia is the only treatment clinically available to reduce the severity of hypoxic-ischemic encephalopathy (HIE) in full-term neonates exposed to HI. This therapeutic strategy has several limitations because it only provides a partial amount of neuroprotection, has a relatively a short time window of therapeutic efficacy, and can only currently be used in neonates born at full term [3,4,5,6,7]. Treatment with therapeutic hypothermia also requires relatively sophisticated technology to induce continuous hypothermia for 72 h and to appropriately monitor the infants. These therapeutic strategies might not be readily available in low socioeconomic environments, or easily implemented in rural or emergency settings [8]. Consequently, studies have questioned the feasibility, safety and efficacy of the use of therapeutic hypothermic treatment for neonates in low-resource settings [8]. The identification of accessible and novel efficacious alternative treatment strategies that could be easily stored, administered and relocated when required could provide additional benefits to limit brain injury and improve behavioral outcomes in neonates exposed to HI.

Inter-alpha inhibitor proteins (IAIPs) are a family of proteoglycans that have immunomodulatory functions. IAIPs are endogenously present in the circulation and in many cell types [9,10,11,12,13], where they act to inhibit destructive proteases, decrease the quantity of damaging cytokines, and alter immune cell responses [14]. Treatment of neonatal rats with exogenous human IAIPs (hIAIPs) after exposure to HI has been shown to decrease infarct volumes in the brain, preserve neurons and oligodendrocytes, decrease the number of microglia, attenuate peripheral immune responses, and improve behavioral outcomes [15,16,17,18,19,20,21]. Therapeutic hIAIPs are isolated and purified from human plasma, which is a costly process and subject to human plasma availability. Therefore, it is important to determine the lowest appropriate dose of hIAIPs that could provide the greatest amount of neuroprotection. This would ensure optimal benefits of this potential therapeutic strategy and limit the amount of hIAIPs needed by administering doses that do not exceed those required for their therapeutic potential.

Treatment with 60 mg/kg of hIAIPs reduced infarct volumes in the brains of neonatal rats by 50% after exposure to 2 h of 8% O_2_, which could be considered as exposure to severe HI [18]. In addition, treatment with 30 mg/kg of hIAIPs reduced infarct volumes in the brains of neonatal rats by 35% after exposure to 90 min of 8% O_2_, which could be considered as exposure to moderate HI [16]. However, these doses of hIAIPs were not investigated under the same experimental paradigms or under the same amount of injury severity. Therefore, evidence is not available to determine whether or not the higher doses provided additional benefits. Several studies have also suggested that there could be differential sex-specific effects of treatment with hIAIPs. These include male but not female improvements in neuronal cell death, pathological scoring of brain tissue, immune responses, and efficacy after delayed administration of hIAIPs [15,16,18]. The reason for the underlying sex-specific effects remains to be determined. It is also possible that differential dosing regimens could be required for male and female neonates to optimize treatment benefits.

In the present study, the efficacy of 30, 60 or 90 mg/kg of hIAIPs was tested when administered intraperitoneally after exposure to HI in neonatal rats. Based upon the considerations summarized above, the hypothesis of the current study is that elevated doses of hIAIP provide superior neuroprotective efficacy for both male and female cohorts of neonatal rats.

## 2. Results

### 2.1. Concentrations of hIAIPs in Rat Serum

The concentration of hIAIPs in rat serum was measured 72 h after exposure to HI, and 24 h after the third injection of intraperitoneal (i.p.) hIAIP or a placebo (PL). As expected, the groups treated with the placebo (phosphate-buffered saline, PBS) rather than hIAIP had low concentrations of measured serum hIAIPs (Figure 1: Sham and HI-PL). Treatment of the HI-exposed neonatal rats with the 30, 60, and 90 mg/kg doses of hIAIPs resulted in serum hIAIP concentrations that were higher (*p* < 0.001) than those in the male and female sham and HI-PL-treated groups (Figure 1). The hIAIP serum concentrations in the female rats were significantly higher in the group treated with 90 mg/kg hIAIPs (125 mg/L) compared with the group treated with 30 mg/kg hIAIPs (Figure 1A: 72 mg/L, *p* < 0.05). The group treated with 60 mg/kg hIAIPs (99 mg/L) had hIAIP serum concentrations that did not differ significantly from the groups treated with 30 or 90 mg/kg of hIAIPs (Figure 1A). The hIAIP serum concentrations in the male neonatal rats did not differ significantly between the groups treated with 30 (67 mg/L), 60 (96 mg/L) or 90 (90 mg/L) mg/kg of hIAIPs (Figure 1B). Therefore, hIAIPs successfully reached the systemic circulation after i.p. administration and remained detectable 24 h after the final injection. Although the full pharmacokinetic profile at multiple time-points would be required to determine hIAIP exposure over the entire 72 h treatment period, the data from the female rats indicated that increasing the dose of hIAIPs administered from 30 to 90 mg/kg was associated with increases in the serum levels. However, the 1.4-fold increase observed in the serum concentrations at the time-point analyzed (Figure 1A) was not proportionally linear to the 3-fold increase in the dose of hIAIP administered.

### 2.2. Body and Brain Weight

The average initial body weight for all rat pups in the study was 17 ± 1 g. There were no significant differences in the initial weights between any of the 10 groups, which were separated by sex and treatment (data not shown).

The changes in body weights of the pups at 24, 48 or 72 h after exposure to HI are shown in Figure 2A,B. The sham group was significantly (*p* < 0.05) heavier than the HI-PL-, HI 30, HI 60, and HI 90 mg/kg hIAIP-treated groups at 24, 48 and 72 h after exposure to HI (Figure 2A,B). Groups exposed to HI did not gain weight from 0 to 24 h, before returning to a trajectory in daily weight gain that was similar to the sham groups from 24 to 72 h (Figure 2A,B). Body weights did not differ significantly between the groups treated with 30, 60, or 90 mg/kg hIAIPs at any time-point for the females or males (Figure 2A,B). Therefore, treatment with hIAIPs (30–90 mg/kg) did not significantly attenuate the reductions in body weight gain over the first 24 h after exposure to severe HI.

Brain weights were significantly (*p* < 0.05) higher in the female sham-treated group compared with the HI-PL- (0.83-fold), HI 30 (0.91-fold), HI 60 (0.89-fold) and HI 90 (0.89-fold) mg/kg hIAIP-treated groups. There were no significant differences in brain weight between the HI-PL- and the HI 30, HI 60, or HI 90 mg/kg hIAIP-treated groups (Figure 2C). Therefore, treatment with hIAIPs (30, 60, or 90 mg/kg) did not significantly attenuate the reductions in female brain weight after exposure of neonatal rats to severe HI. Brain weights were not significantly different between male sham and HI-PL groups (Figure 2D).

### 2.3. Behavioral Assessment

#### 2.3.1. Righting Reflex

The time taken to complete a righting reflex task is shown in Figure 3. The female HI-PL group took significantly longer to complete the reflex task than the sham group (Figure 3A, 1.9-fold, *p* < 0.05). The HI 30 (0.9-fold), HI 60 (1.2-fold) and HI 90 (1.0-fold) mg/kg hIAIP-treated groups did not have significantly different times to complete the reflex compared to the sham group (Figure 3A). Therefore, treatment of female rats with hIAIPs (30, 60, and 90 mg/kg) after neonatal exposure to HI attenuated righting reflex speed to times that did not differ significantly from the sham group. The time to complete the reflex was not significantly different between male sham and HI-PL groups (Figure 3B). Therefore, the test could not distinguish differences between the HI-PL- and HI-hIAIP-treated groups of male neonatal rats.

#### 2.3.2. Negative Geotaxis

There were no significant differences between any groups for the time taken to turn 180° to face the inclined plane, indicating that this test was not able to determine behavioral differences after neonatal HI brain injury in females or males (Appendix A). Therefore, the test could not distinguish differences between the HI-PL- and HI-hIAIP-treated groups of neonatal rats.

#### 2.3.3. Small Open-Field Task

Significant differences were not detected between the sham and HI-PL groups for distance travelled, maximum acceleration, average velocity, rotations, time in the outer twelve segments or time in the center four segments during the small open-field 10 min exploration task (Appendix A).

The percentage of zones entered during exploration of the open-field is shown in Figure 4A,B for the females and males, respectively. The majority of pups in the sham group entered all zones (8/10 females; 9/10 males), with an average of 94% of zones entered in the male and female sham groups (Figure 4A,B). The HI-PL group of females explored significantly fewer zones than the sham group (Figure 4A, 0.7-fold, *p* < 0.05). There were no significant differences in the percentage of zones entered between the sham group of females and the HI 30, HI 60, or HI 90 mg/kg hIAIP-treated groups (Figure 4A). The female HI 60 mg/kg hIAIP-treated group entered significantly more zones than the HI-PL group (Figure 4A). Therefore, treatment with hIAIPs (30, 60, 90 mg/kg) after exposure of the female neonatal rats to HI attenuated zone exploration to levels that did not differ significantly from the sham group. The percentage of zones entered was not significantly different between male sham and HI-PL groups (Figure 4B); therefore, this task could not distinguish differences between the HI-PL- and HI-hIAIP-treated groups of male neonatal rats.

The number of times that elongated or contracted postures were exhibited during the small open-field task was significantly higher in the HI-PL group of male neonatal rats compared to the sham group (Figure 4D, 1.8-fold, *p* < 0.05). There were no significant differences in the number of postural changes between the sham group of males and the groups treated with 30, 60, or 90 mg/kg hIAIPs (Figure 4D). The male HI 60 (0.61-fold, *p* < 0.05) and HI 90 (0.50-fold, *p* < 0.01) mg/kg hIAIP-treated groups exhibited significantly fewer extension and contraction postures than the HI-PL group (Figure 4D). Therefore, treatment with hIAIPs (30, 60, 90 mg/kg) after exposure to HI in the male neonatal rats significantly reduced the number of postural changes in the small open-field task to frequencies that did not differ significantly from the sham group. The HI 90 mg/kg hIAIP-treated group exhibited the greatest fold reductions in the number of extension and contractions postures exhibited compared to the HI-PL group (Figure 4D, 0.50-fold, *p* < 0.01). There were no significant differences in the number of extension and contractions postures exhibited between the female sham group and the HI-PL group (Figure 4C). Therefore, this task could not distinguish differences between the HI-PL- and HI-hIAIP-treated groups of female neonatal rats.

### 2.4. Brain Infarct Volumes

The average whole hemisphere infarct volumes in the HI-PL group were 71% in the males and 65% in the females, the average cortical infarct volumes were 80% in the males and 77% in the females and the average striatal infarct volumes were 91% in the males and 79% in the females (Figure 5 and Figure 6). These quantities of infarct volumes suggest a severe degree of brain injury for the Rice–Vannucci model. Representative images of infarct volumes in the placebo-treated males and females are shown in Figure 7. There were no significant differences in the extent of the infarct volumes in the HI-PL group compared with the HI 30, HI 60, or HI 90 mg/kg hIAIP-treated groups in the whole hemisphere or cortex in the males (Figure 5A,B), or any of the brain regions in the female neonatal rats (Figure 6). There were significantly lower infarct volumes in the striatum of the HI 30 (0.74-fold, *p* < 0.05), HI 60 (0.74-fold, *p* < 0.05) and HI 90 (0.74-fold, *p* < 0.05) mg/kg hIAIP-treated groups compared with the HI-PL-treated group of males (Figure 5C). Significant differences were not observed between the infarct volumes of HI 30, HI 60, or HI 90 mg/kg hIAIP-treated groups for either sex in any brain region. Therefore, all animals in the HI 30, HI 60, and HI 90 mg/kg hIAIP-treated groups were combined into one hIAIP group per sex for subsequent analysis (Figure 8 and Figure 9). The combined HI-hIAIP (30, 60, 90 mg/kg) group of males exhibited significantly lower infarct volumes than the HI-PL group in the hemisphere (0.75-fold, *p* < 0.05), cortex (0.77-fold, *p* < 0.05) and striatum (0.74-fold, *p* < 0.05; Figure 8B). Significant differences were not observed in the females (Figure 8A).

Significant correlations were not observed between terminal hIAIP concentrations (Figure 1) and either infarct volumes in any brain region or performance in the behavioral tests (data not shown). Spearman correlation analysis identified significant correlations between the total number of demonstrations of contracted or extended postures during open-field exploration and infarct volumes of the whole hemisphere (R = 0.46, *n* = 30, *p* < 0.05), cortex (R = 0.38, *n* = 30, *p* < 0.05) and striatum (R = 0.48, *n* = 30, *p* < 0.01) brain regions (Figure 9) in the males exposed to severe HI and treated with hIAIPs (30, 60, 90 mg/kg). The correlations were not significant between brain infarct volumes and performance in the righting reflex or open-field exploration tasks in the HI-hIAIP treated females (data not shown). Therefore, treatment with hIAIPs after exposure of neonatal females to HI may result in preservation of behavioral capacities through mechanisms not directly related to the protection of gross brain volume.

The number needed to treat (NNT) to prevent an incidence of >60%, >50%, >40%, >30%, and >20% infarct volume in the hemisphere was 5, 8, 8, 8 and 15, respectively, for females and 3, 3, 3, 4 and 6, respectively, for males.

## 3. Discussion

HI-related brain injury in neonates can result in death or irreversible neurological damage that predisposes the infant to the development of a range of neurodevelopmental and cognitive disorders [22,23]. The current standard of care to attenuate brain injury in full-term infants exposed to HIE is therapeutic hypothermia for 72 h. However, therapeutic hypothermia has some limitations because it is not approved for use in prematurely born infants, has a relatively a short time window of therapeutic efficacy after birth, its neuroprotective effects are often limited to partial protection in the infants with HIE, and it is more effective after exposure to moderate, rather than severe, HIE [3,4,5,6,7,8]. In addition, injectable agents might be more feasible to treat neonates in low socioeconomic, rural or emergency settings, in which access to multiple portable sets of hypothermic therapeutic technologies, monitoring equipment and the relevant trained personnel could be difficult. Consequently, there is a need for novel therapeutic strategies that could be administered either as alternative or adjunctive treatments to therapeutic hypothermia [24,25].

hIAIPs are a novel therapeutic candidate that has been shown to have important neuroprotective properties in preclinical rodent studies [14,15,16,17,18,19,20,21,26]. The optimal treatment regimen for hIAIPs needs to be ascertained before consideration of a potential clinical trial, in order to maximize their beneficial neuroprotective properties. Producing IAIPs with recombinant techniques would be challenging because they are endogenously transcribed by four different genes localized on three different chromosomes [27,28,29,30,31]. Consequently, IAIPs are extracted and purified from human plasma, as described below in the section “Preparation of hIAIPs” [15,16,18,21,26,32]. Treatment with hIAIPs effectively reached the systemic circulation after the i.p. injections and was present for up to 24 h after the final dose of hIAIPs (Figure 1). Previous pharmacokinetic analysis has shown that hIAIP concentrations reach their highest concentrations in blood 1–12 h after an i.p. injection, and have a half-life of 10–23 h [32]. Complete pharmacokinetic analysis would be required to identify whether increasing the dose of hIAIPs administered results in increased peak or total hIAIP concentrations in serum. The results available from the present study demonstrate that increasing hIAIP doses from 30 to 90 mg/kg/day can result in increased drug concentrations in the blood of females (Figure 1A). However, the 3-fold increase in the dose only resulted in a 1.4-fold increase in hIAIP serum concentrations 24 h after the last dose of hIAIPs. We cannot comment upon the hIAIP concentrations in blood shortly after administration because the samples were obtained after the termination of the studies. Nonetheless, it remains probable that there is a maximal limit to the amount of hIAIPs that can be absorbed from the peritoneum into the bloodstream. In addition, it is potentially feasible that the drug clearance or redistribution from blood into other organs and tissues results in a highly regulated level of drug in blood at a given time. Additional pharmacokinetic profiling of different hIAIP doses would be useful to determine whether increasing the administered hIAIP dose would result in increased drug bioavailability and increased concentrations in blood and brain vasculature.

Previous studies have shown that treatment with hIAIPs can improve behavioral performance in the water maze [17,19], on auditory discrimination [20], and wire hang tasks [21]. The righting reflex requires a combination of sensory perception for the rats to recognize that they are in a supine position, the innate cognitive capacity to realize that they prefer to be in a prone position, and motor function to return to the prone position. The female, but not male, pups took an increased amount of time to return to the prone position after exposure to severe HI compared with the sham group. Treatment with doses of 30 mg/kg or greater were sufficient to attenuate this deficit (Figure 3A). These findings are somewhat consistent with our previous work, in which hIAIPs at a dose of 30 mg/kg attenuated deficits in the righting reflex after exposure to a more moderate amount of HI in male, but not female, neonatal rats [21]. Combined, the results from the present study and our previous publication suggest that restitution of righting reflex behaviors by treatment with hIAIPs can be observed in both males and females. However, it appears that for this behavioral test, only the females exhibited increased time to complete the reflex after exposure to severe HI (Figure 3). The reason that the males did not exhibit deficits in the righting reflex task in the present study cannot be discerned from our current work. However, it should be pointed out that the study design in our previous work differed from the present study because the model of injury exhibited less severity and the righting reflex test had been sequentially performed over a three-day interval [21].

The female, but not the male pups, entered significantly fewer zones in the open-field task after exposure to severe HI, compared with the sham group (Figure 4). This is more likely a result of a decreased perception of safety or decreased interest in exploration, rather than a decrease in motor capacity because there were no significant differences in the total distance traveled between the HI-PL and sham groups. Our previous work has shown significant differences in the distance traveled between the placebo-treated HI-exposed and sham groups at P13, six days after exposure to moderate HI, which was attenuated by treatment with hIAIPs [21]. This suggests that although we did not identify differences in the total distance travelled in the open-field task between the HI-PL and sham groups at P10, differences might have been detected if the duration of the experiment had been extended. Nonetheless, females treated with 60 mg/kg of hIAIPs explored a significantly greater number of zones than the HI-PL group (Figure 4A). There were no significant differences between zones entered by the groups treated with each of the three individual doses of hIAIPs, all of which did not differ significantly from the sham group. Exploration of the open-field domains in the male animals was not affected by exposure to HI (Figure 4B). However, the male neonatal rats exposed to HI exhibited a greater number of extension and contraction postures compared with the sham controls (Figure 4D). High-frequency postural changes to extended or contracted positions can be a sign of anxiety or distress [33,34], signifying the investigation of, or reactions to, perceived threats, despite their absence. Treatment with 30, 60, and 90 mg/kg hIAIPs attenuated the increased frequency of the postural changes to levels that did not differ from the sham-treated group (Figure 4D). It should be noted that only groups treated with 60 and 90 mg/kg hIAIPs had significantly fewer postural changes compared to the HI-PL group, which may provide some evidence to support the potential benefits of increasing the dosage above 30 mg/kg (Figure 4D). However, because no significant differences were identified between the groups treated with 30, 60 or 90 mg/kg hIAIPs, or the sham group, the results from the present study cannot conclude that 30 mg/kg treatment was not sufficient to achieve treatment benefits (Figure 4). The possibility of dose-dependent effects of hIAIPs on behavioral recovery after exposure to HI should not be disregarded, particularly for alternate behavioral analyses, or in injury models of lower severity.

Overall, the present study demonstrates that exposure to severe HI predisposes male and female neonatal pups to impaired function in some, but not all, behavioral domains. The sex-specific behavioral deficits observed may relate to the known differences in the mechanism(s) and outcomes of HI-related brain injury between male and female neonates [35], which continues to evolve as an important area of preclinical and clinical research. The variations in the types of behavioral deficits exhibited in the male and female neonatal rats after exposure to HI cannot be explained by the current study, but emphasize the need to examine both males and females separately after exposure to HI. Nonetheless, hIAIPs modified the responses to all of the abnormalities in the behavioral domains identified in both males and females in the present study.

The HI model employed in this study resulted in severe brain injury, with damage in approximately 70% of the right hemisphere, 80% of the right cortex and 85% of the right striatum (Figure 5 and Figure 6). Although all of the behavioral abnormalities observed were attenuated by treatment with hIAIPs, the infarct volume of damaged brain tissue remained extensive in many of the hIAIP-treated male and female animals (Figure 5 and Figure 6). The exposure to HI in this study was based upon our former work, in which hIAIPs exerted a significant protective effect after exposure to carotid artery ligation and 8% oxygen for 120 min [18]. However, the neonatal rats in this study were maintained at 37 °C during hypoxia, rather than at 36 °C as in our former work [18], which could, in part, account for the more severe injury in the current study compared with our former work [18]. Maintenance of body temperature at 37 °C instead of 36 °C was selected based on our former behavioral studies to ensure differences between the HI-PL and sham groups [16,18,21]. However, these temperature differences could have contributed to a severity of injury that limited the capacity to observe hIAIP-related beneficial treatment effects on the infarct volumes. Even small increases in body temperature during HIE can increase the severity of brain injury in human infants [36]. Although treatment with hIAIPs did not significantly reduce the infarct volumes of the hemisphere or cortex in the male neonatal rats or the any of the brain regions in the female neonatal rats after exposure to severe HI, the 30, 60, and 90 mg/kg doses of hIAIPs significantly reduced the infarct volumes to an equivalent extent in the striatum of the male neonatal rats (Figure 5C). The lack of statistically significant effects of hIAIPs on the infarct volumes in the hemisphere and cortex in the males and the three brain regions in the females could be due, in part, to the large variability in treatment effects observed in the hIAIP-treated groups.

Two different patterns of tissue injury were observed after treatment with hIAIP in both the males and females (Figure 5, Figure 6, Figure 7 and Figure 8). Results from the present study (e.g., treatment group, terminal hIAIP levels, starting weight etc.) did not correlate with the infarct volumes in a manner that could predict or explain why some animals exhibited greater responses to the treatment compared with others. Our previous work also demonstrated similar patterns of the responders and non-responders after treatments to hIAIPs, resulting in bimodal distributions similar to the current study [18,21]. This could, in part, be a result of the well-known variability in the Rice–Vannucci HI model of brain injury [37,38,39,40]. A large retrospective study that analyzed over 1000 neonatal rats with Rice–Vannucci HI-induced brain injuries also reported a high variability in the treatment effect of therapeutic hypothermia, when measuring infarct volumes in the brain [41]. The study identified a bimodal distribution of the treatment effect, similar to those observed in the present study, suggesting that although analyzing very large group numbers would not decrease variability in the datasets, increased numbers could be required to allow for the level of statistical power required to detect the effects of novel treatments in this model of brain injury [41]. Therefore, future studies that analyze hIAIP treatment efficacy may need to consider larger animal numbers in order to identify all of the aspects of neuroprotection that could translate into clinical benefits.

Significant differences in the infarct volume measurements of the groups treated with 30, 60 or 90 mg/kg doses of hIAIPs were not observed. Consequently, increases in the dose of hIAIPs from 30 mg/kg to either 60 or 90 mg/kg did not appear to alter the average infarct volumes for either males or females in the injury model examined (Figure 5 and Figure 6). Therefore, we combined the three hIAIP-treated groups into one hIAIP treatment group (Figure 8) for the infarct volume analysis and confirmed that treatment with hIAIPs reduced the infarct volumes in the hemisphere, cortex, and striatum of the male neonatal rats, but not the female neonatal rats, after exposure to severe HI. In this context, it is important to emphasize that hypothermia was not neuroprotective in previous studies when applied to neonatal rats after exposure to severe HI [39,41,42]. The severity of the HI insults and the 37 °C temperature used during the induction of HI were similar in the present study and the former work [39]. Therefore, even the relatively modest amount of neuroprotection afforded by hIAIPs in male cohorts is important, with respect to the lack of effect by hypothermia after severe HI in the same model [39]. In addition, previous studies have reported that hypothermic treatment can provide a significant attenuation of brain infarct volumes in female, but not male, rats exposed to moderate HI [41] and that it also attenuates behavioral deficits in a sex-specific manner [43]. Although the clinical relevance of these results remains to be determined [44], the sex-specific differences in efficacy between hIAIPs and hypothermia in some preclinical testing of neuroprotection suggest that direct comparisons between the two treatments in future studies may be important to identify optimal treatment options for males and females.

In the present study, the NNT to reduce infarct volumes in the brain to below 30% of the hemisphere was eight for females and four for males. Hypothermia treatment has been reported to have an NNT of 10 to achieve equivalent infarct parameters in a similar preclinical rodent model of injury [42]. Treatment with therapeutic hypothermia after exposure of full-term infants to HIE has been reported to have an NNT of 7–14 over a range of mortalities and neurodevelopmental parameters [6,45]. The possibility that hIAIPs could provide adjunctive benefits to the standard treatment with therapeutic hypothermia needs be considered in future studies. Additional studies that examine the efficacy of hIAIPs with and without exposure to therapeutic hypothermia could be further used to determine its efficacy in fetal sheep and non-human primates. The quantification of cellular subtypes in both the hypoxic and hypoxic-ischemic hemispheres of the brain within these studies may assist in differentiating neuroprotective efficacy.

The present study identified novel sex-specific behavioral deficits for male and female rats after exposure to severe HI. The attenuation of the behavioral deficits observed following the 30, 60, or 90 mg/kg doses of hIAIPs supports the protective capacity of this novel therapeutic strategy for both males and females. Histological analysis also identified the neuroprotection provided by hIAIPs in the males by reducing the infarct volumes in a manner that correlated with the behavioral improvement. Nonetheless, the 30 mg/kg dose was as efficacious as the 60 and 90 mg/kg doses. Therefore, the original hypothesis that higher doses of hIAIPs would prove to be more efficacious than the lower doses to treat HI-related brain injury after exposure to HI cannot be supported by the findings of the current study. However, the paradigm in the current study resulted in a very severe degree of injury to the brain. Consequently, we cannot rule out the possibility that an effective graded dose-response effect could have been detected if the neonatal rats had been exposed to a more moderate degree of HI. The beneficial effects on the behavioral outcomes and infarct volumes in the brain provided by hIAIPs, when administered after severe HI, could be important because the current recommended hypothermia regimen is more effective in treating infants with moderate, rather than severe, HIE [39,45].

## 4. Materials and Methods

### 4.1. Animals

The Institutional Animal Care and Use Committees of the Alpert Medical School of Brown University and Women & Infants Hospital of Rhode Island approved the animal experimentation in this study. All procedures were conducted in accordance with the National Institutes of Health Guidelines for the Use of Experimental Animals. Pregnant wild-type Wistar *Rattus norvegicus* were obtained from Charles River Laboratories (Wilmington, MA, USA) on embryonic day 15–16. Rats were housed in a temperature-controlled environment with a 12 h light/dark cycle and had ad libitum access to food and water. The pups were housed in a cage with the dam and littermates from birth, until the conclusion of the study.

Animals were randomly assigned to one of the following five groups: (i) sham placebo- (sham), (ii) HI placebo- (HI-PL), (iii) HI 30 mg/kg hIAIP- (HI-30), (iv) HI 60 mg/kg hIAIP- (HI-60) or (v) HI 90 mg/kg hIAIP (HI-90)-treated groups. Ten pups per sex were allocated to each group to ensure random allocation of *n* = 10/sex/group. The male HI-PL group had *n* = 9 after 1 animal was removed from all analyses, due to an incomplete carotid artery occlusion that had been noted during surgery. The 30 and 60 mg/kg doses were selected based upon our previous studies [15,16,17,18,19,20]. The 90 mg/kg dose represented the maximum single i.p. injection that was feasible with the concentration of the hIAIPs currently available. Sham hIAIP-treated groups were not considered for analysis due to the lack of significant differences observed in previous studies compared to the sham placebo groups for the infarct volumes previously measured [18].

### 4.2. Surgeries and Study Design

On postnatal day 7 (P7), the rat pups underwent either sham surgery or HI surgery using the Rice–Vannucci unilateral occlusion model, as previously described [46]. All animals were anesthetized with isoflurane in oxygen (4% induction, 2% maintenance) and a small incision was made in the neck. The right carotid artery was ligated in the HI groups. An equivalent incision was made in the sham group, but the right common carotid artery was not ligated. Rat pups were returned to the dam for feeding and allowed to recover from surgery for 90 min. Rats in the HI groups were placed in a hypoxia chamber (8% O_2_, 92% nitrogen: Biospherix, Parish, NY, USA) and the rats in the sham group were placed in an atmospheric chamber for 2 h. Rectal temperature (RET-4, Physitemp, Clifton, NJ, USA) was maintained at 37 °C throughout the 2 h period, which was measured in a sentinel rat pup that was not used for further analysis [47,48]. This temperature was selected to ensure maximal injury for the behavioral assessments of the treatment effects of hIAIPs [19,20] and is slightly higher than in some of our previous reports of biomolecular and histopathological efficacy of treatment with hIAIPs [15,16,18,21]. hIAIPs were administered intraperitoneally immediately after hypoxia at 0 h and repeated at 24 and 48 h after exposure to HI (Figure 10). Phosphate-buffered saline (PBS) was administered as the placebo because it was used as the vehicle for hIAIPs. The hIAIPs used in this study were extracted from freshly frozen human plasma (Rhode Island Blood Center, RI, USA). The same batch of hIAIPs was used for all treated subjects. The study design is illustrated in Figure 10.

### 4.3. Preparation of hIAIPs

hIAIPs were prepared as previously described [18,19] with the same purification methodology that was used in the previous studies [16,21]. All animals were treated with the identical batch of hIAIP. hIAIPs were extracted from human plasma (Rhode Island Blood Center, RI, USA) and purified using anion-exchange chromatography on Toyopearl GigaCap Q-650M (Tosoh Bioscience, King of Prussia, PA, USA). Subsequently, the bound proteins were eluted and further subjected to synthetic chemical ligand affinity chromatographic media (Astrea Bioseparations, Cambridge, UK). Eluted proteins were concentrated and buffer exchanged using a tangential flow filtration device (Labscale TFF System, MilliporeSigma, Burlington, MA, USA), containing a Pellicon XL50 cartridge with a 30 kDa Biomax membrane (MilliporeSigma, Burlington, MA, USA). Sample purity was confirmed using SDS-PAGE, Western immunoblot, protein assay, and competitive immunoassay [49]. Biological activity was measured using a spectrophotometry assay based on the ability of IAIPs to inhibit the hydrolysis of N-benzoyl-L-arginine-p-nitroaniline HCl (L-BAPNA, Millipore Sigma, St. Louis, MO, USA) by trypsin, resulting in decreased rates of change in absorbance per minute (405 nm) [50].

### 4.4. Behavioral Analyses

#### 4.4.1. Righting Reflex

The righting reflex task was performed on P8, 23 h after exposure to HI or sham treatment and before treatment with the second dose of hIAIPs or the placebo. The pups were placed on their backs on a flat surface and were given the opportunity to right themselves to a prone position on their paws. The time to right themselves was recorded when the pup had placed all four paws flat on the surface in a position that could support ambulation. If their paws were trapped under their body, excessively sprawled, or turned so that the flat parts of their feet faced upwards, timing was continued until the final position was achieved. A maximum of 20 s was permitted for completion of the task. All attempts that were not completed were added to the maximum score. Each pup completed the task 5 times. All experiments were video-recorded and analyzed by three independent observers that were not aware of the group designations of each animal.

#### 4.4.2. Negative Geotaxis

Negative geotaxis performance was examined on P9, 47 h after exposure to HI or sham treatment and before exposure to the third dose of hIAIPs or the placebo. The pups were placed in the center of a 30° inclined wire mesh (26 × 20 cm), facing downwards. The experimenter removed their hands once the pup grabbed the wire mesh sufficiently to support its own body weight, thereby permitting the animal to turn to face upwards on the inclined platform. The period taken for the vertical plane of the pup to turn 180° to face upwards on the platform was timed. Each pup completed the task three times. All experiments were video-recorded and analyzed using EthoVision XT (Noldus, Waginen, The Netherlands) tracking software.

#### 4.4.3. Small Open-Field Tasks

Small open-field tasks were conducted on P10, 71 h after exposure to HI and before humane killing and brain sampling at 72 h. Pups were placed in the center of an open-field box (28 × 28 cm) and allowed to explore for 10 min. All experiments were video-recorded and analyzed using EthoVision XT (Noldus, Waginen, The Netherlands) tracking software. The arena was segmented into 16 even square segments for analysis of zone entry and time spent in the outer and center zones of the arena. The endpoints analyzed were distance traveled, velocity, time in the outer 12 segments, time in the center 4 segments, number of zones entered, acceleration, rotation (clockwise; anticlockwise) and body elongation state (contracted, normal, and stretched).

### 4.5. Sample Collection

Animals were humanely killed 72 h after exposure to HI or sham treatment. Blood was sampled via cardiac puncture, which was allowed to coagulate and was centrifuged (1200 g, 5 min) to acquire the serum. Serum samples were stored at −80 °C until analysis. Brains were perfused with PBS, paraformaldehyde (PFA, 4%) and then weighed. Brains were placed in 4% PFA for 24 h, then stored in phosphate-buffered sucrose solution (30%). Brains were sectioned into 2 mm coronal segments using a brain slicer matrix (Zivic instruments, Pittsburgh, PA, USA), then each segment was placed in a plastic mold, covered in optimal cutting temperature compound (OCT) and frozen by placing the mold into isopentane on a bed of dry ice. Frozen brains were stored at −80 °C, until sectioned on a cryostat (CM3050S, Leica, Nusslock, Germany) at 20 μm and mounted on gelatin-coated slides.

### 4.6. Histology

Every seventh cryosection per 2 mm brain segment was selected for histological quantification. The number seven was randomly decided. The slides were removed from the −80 °C freezer and air-dried overnight. Slides were then exposed to 1:1 chloroform and absolute ethanol (20 min). Sections were each stained with cresyl violet (0.1% *w*/*v*, 6 min; MilliporeSigma, St. Louis, MO, USA), then rinsed with dH_2_O (1 min). The sections were then subjected to 4 min washes of 95% ethanol, 100% ethanol, and 95% ethanol. The slides were then air-dried overnight and mounted (Cytoseal XYL, Richard-Allan Scientific, Canton, MI, USA). Images of each section were obtained using a Micropublisher 6 CCD Camera (QImaging, Surrey, BC, Canada). Stained tissue volume was measured with ImageJ by two independent observers that were not aware of the group assignments. Total stained tissue, stained cortex, and stained striatum were quantified for the left and right hemispheres of each brain section (Figure 7). Stained tissue volumes were calculated using the known 2 mm distances between each section. Total infarct volume was examined using the following formula: infarct (%) = [1 − (ipsilateral volume/contralateral volume)] ×100 [18,51]. Negative values were adjusted to 0 to signify no infarct.

### 4.7. ELISA of hIAIP Concentrations in Rat Serum

The levels of hIAIP were measured quantitatively by competitive enzyme-linked immunosorbent assay (ELISA), using an immobilized monoclonal antibody against the light chain of IAIP (MAb 69.26) and a biotinylated purified hIAIP as the detecting reagent. IgG of MAb 69.26 purified from the hybridoma cell culture media was immobilized on Microlon 600 96-well high-binding microplates (Greiner Bio-One, Monroe, NC, USA). The ELISA was performed as follows: a 200 ng/well of purified IgG of MAb 69.26 was diluted in PBS and bound to a 96-well plate for 1 h at room temperature (RT). This was followed by blocking with 2% BSA in PBS for 1 h at RT. The plasma samples were diluted (1:100) in PBS and incubated for 1 h at RT, along with the biotinylated purified human IAIP (diluted 1:1000). After extensive washing, horse-radish peroxidase-conjugated streptavidin (Millipore Sigma, St. Louis, MO, USA) was added and the plate was incubated for 30 min at RT. Then, 100 μL of the chromogenic substrate solution Enhanced K-Blue TMB (Neogen, Lexington, KY, USA) was added to the well after washing, and after 20 min, 1 M of hydrochloric acid solution was added to stop the reaction. Absorbance was read using the spectrophotometer SpectraMax Plus (Molecular Devices, San Jose, CA, USA) at 450 nm wavelength. Each plasma sample was tested in triplicate and the mean value was used to calculate the hIAIP concentration against the values of the known hIAIP standard.

### 4.8. Statistics

Statistical analysis was selected based upon the best fit for the data in each individual experiment. The figure legends show the statistical tests used for each dataset. In short, the data were analyzed using a generalized linear model, the Kruskal–Wallis test, negative binomial model, maximum likelihood testing or proportional hazards modeling. Tukey–Kramer correction was used to adjust for multiple comparisons. Infarct volume measurements of multiple brain regions from cresyl violet slides were analyzed using two-factor analysis of variance for repeated measures. Spearman nonparametric testing was used for correlation statistics. Adjusted *p* values below 0.05 were considered statistically significant for all the analyses. Statistical analysis and graphical representations were conducted using SAS version 9.4 (Cary, NC, USA) and GraphPad Prism (Sand Diego, CA, USA).

## Figures and Tables

**Figure 1 ijms-23-13473-f001:**
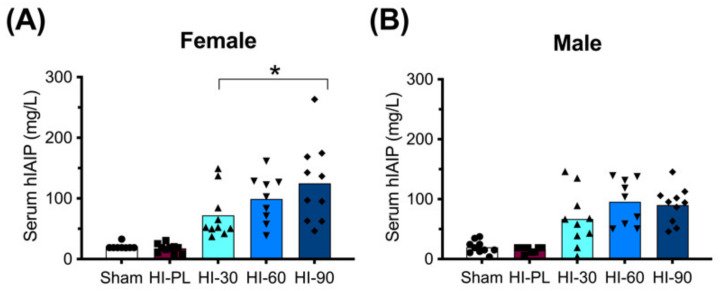
Human inter-alpha inhibitor protein (hIAIP) concentrations in the serum of rats 72 h after hypoxia–ischemia (HI) and injections at 0, 24 and 48 h. Results are shown for the sham placebo- (sham, open bar, circles), HI placebo- (HI-PL, red bar, squares), HI 30 mg/kg hIAIP- (HI-30, light blue bar, upward-facing triangles), HI 60 mg/kg hIAIP- (HI-60, medium blue bar, downward-facing triangles) and HI 90 mg/kg hIAIP- (HI-90, dark blue bar-diamonds) treated groups. Measurements were made with competitive ELISA. Results are shown for females (**A**) and males (**B**). Significant differences between groups were measured using a negative binomial model with the Tukey–Kramer adjustment for multiple comparisons. Significant differences between HI 30 mg/kg, HI 60 mg/kg and HI 90 mg/kg are shown graphically (* *p* < 0.05). Significant differences between the sham or HI-PL groups and the HI 30, HI 60 or HI 90 mg/kg hIAIP-treated groups were identified for all comparisons (negative binomial model, Wald chi-square for group (4df) 158.62, *p* < 0.0001), but are not shown graphically. Each animal is presented as an individual datapoint.

**Figure 2 ijms-23-13473-f002:**
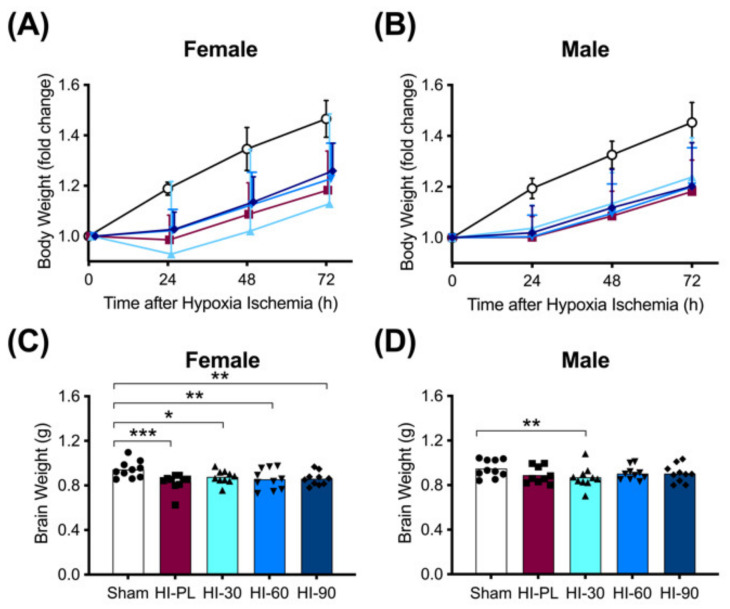
Body (A–B) and brain (C–D) weights after exposure to hypoxia–ischemia (HI) and treatment with human inter-alpha inhibitor proteins (hIAIPs). Results are shown for female (**A**,**C**) and male (**B**,**D**) rats. (**A**,**B**)) Groups are sham placebo- (sham, open circles), HI placebo- (HI-PL, red squares), HI 30 mg/kg hIAIP- (HI-30, light blue triangles), HI 60 mg/kg hIAIP- (HI-60, medium blue downward triangles) and HI 90 mg/kg hIAIP (HI-90, dark blue diamonds)-treated groups. Body weights are fold changes from starting weight (0 h) at 24, 48 and 72 h, presented as means ± standard deviation (*n* = 9–10). (**C**,**D**)) Groups are sham placebo- (sham, open bars, circles), HI placebo- (HI-PL, red bars, squares), HI 30 mg/kg hIAIP- (HI-30, light blue bars, triangles), HI 60 mg/kg hIAIP- (HI-60, medium blue bars, downward triangles) and HI 90 mg/kg hIAIP- (HI-90, dark blue bars, diamonds) treated groups. Brain weights (g) were recorded 72 h after HI; each animal is presented as an individual datapoint. For statistical analysis, we used the generalized linear model with Tukey–Kramer adjustment for multiple comparisons, female Wald chi-square for group (4df) 21.25, *p* < 0.001; male Wald chi-square for group (4df) 11.59, *p* < 0.05, * *p* < 0.05, ** *p* < 0.01; *** *p* < 0.001.

**Figure 3 ijms-23-13473-f003:**
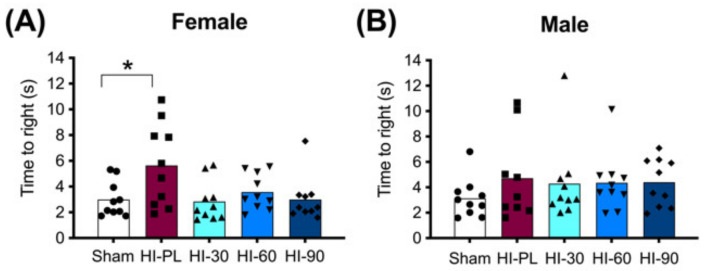
Righting reflex performance on postnatal day 8 (P8), 23 h after exposure to hypoxia–ischemia (HI) and treatment with human inter-alpha inhibitor proteins (hIAIPs). The time(s) taken to complete the righting reflex task by placing for all four paws flat on the bench after the pups were placed on their backs. Results are shown for females (**A**) and males (**B**). Values are an average of five trials per animal and an average of measurements by three observers who were not aware of treatment group allocation. Each animal is presented as an individual datapoint (*n* = 9–10). Groups are sham placebo- (sham, open bars, circles), HI placebo- (HI-PL, red bars, squares), HI 30 mg/kg hIAIP- (HI-30, light blue bars, triangles), HI 60 mg/kg hIAIP- (HI-60, medium blue bars downward triangles) and HI 90 mg/kg hIAIP- (HI-90, dark blue bars, diamonds) treated groups. Significant differences between groups were measured using proportional hazards modeling least squares means adjusted for multiple comparisons with Tukey–Kramer adjustment and Wald chi-square for group (4df) 10.76, *p* < 0.05; * *p* < 0.05.

**Figure 4 ijms-23-13473-f004:**
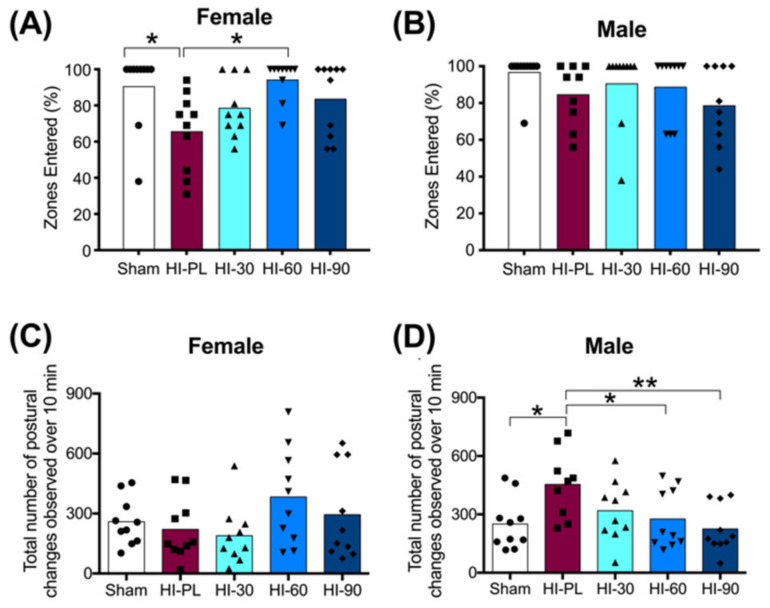
Open-field tasks at postnatal day 10 (P10), 71 h after HI and treatment with human inter-alpha inhibitor proteins (hIAIPs). Groups are sham placebo (sham, open bars, circles), HI placebo (HI-PL, red bars, squares), HI 30 mg/kg hIAIP-treated (HI-30, light blue bars, triangles), HI 60 mg/kg hIAIP-treated (HI-60, medium blue bars, downward triangles) and HI 90 mg/kg hIAIP-treated (HI-90, dark blue bars, diamonds) groups. Results are shown for females (**A**,**C**) and males (**B**,**D**). (**A**,**B**)) The percentage of zones entered by an animal during the 10 min exploration period. (**C**,**D**)) The number of extension or contraction postural changes exhibited during the 10 min exploration period. Each animal is presented as an individual datapoint. Statistical analysis was completed using Kruskal–Wallis test, * *p* < 0.05, z (4df) < −3.1 (**A**,**B**), or generalized linear models (**C**,**D**), with Tukey–Kramer correction for multiple comparisons and Wald chi-square for group (4df) 15.73, *p* < 0.01, * *p* < 0.05; ** *p* < 0.01.

**Figure 5 ijms-23-13473-f005:**
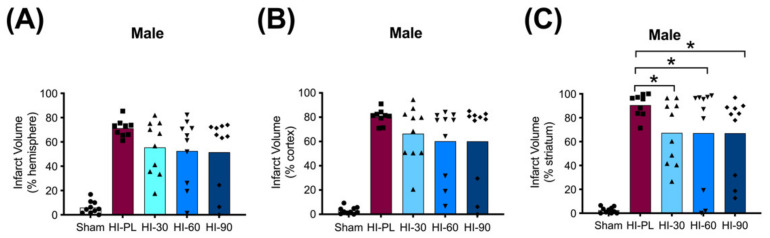
Infarct volumes in the male neonatal rat brains 72 h after exposure to hypoxia–ischemia (HI) and treatment with human inter-alpha inhibitor proteins (hIAIPs). Values are percent infarct volumes measured by cresyl violet staining for sham placebo (sham, open bars, circles), HI placebo (HI-PL, red bars, squares), HI 30 mg/kg hIAIP-treated (HI-30, light blue bars, triangles), HI-60 mg/kg hIAIP-treated (HI-60, medium blue bars, downward triangles) and HI 90 mg/kg hIAIP-treated (HI-90, dark blue bars, diamonds) groups. Infarct volumes for right hemisphere (**A**), right cortex (**B**) and right striatum (**C**) are shown. Each animal is presented as an individual datapoint. Statistical analysis included two-factor analysis of variance for repeated measures, F(4132) = 43.19, * *p* < 0.05. Significant differences between the sham group and other groups are not shown graphically; all significance values between the sham and HI-PL-, HI 30, HI 60, or HI 90 mg/kg hIAIP-treated groups were *p* < 0.05.

**Figure 6 ijms-23-13473-f006:**
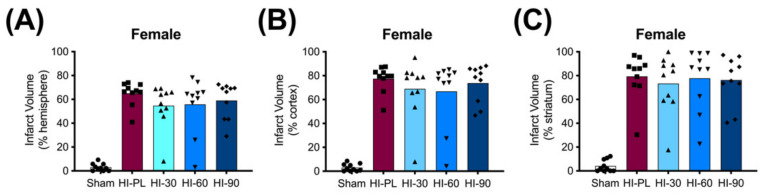
Infarct volumes in the female neonatal rat brain 72 h after exposure to hypoxia–ischemia (HI) and treatment with human inter-alpha inhibitor proteins (hIAIPs). Values are percent infarct volumes measured by cresyl violet staining for sham placebo (sham, open bars, circles), HI placebo (HI-PL, red bars, squares), HI 30 mg/kg hIAIP-treated (HI-30, light blue bars, triangles), HI 60 mg/kg hIAIP-treated (HI-60, medium blue bars, downward triangles) and HI 90 mg/kg hIAIP-treated (HI-90, dark blue bars, diamonds) groups. Infarct volumes for right hemisphere (**A**), right cortex (**B**) and right striatum (**C**) are shown. Each animal is presented as an individual datapoint. Statistical analysis included two-factor analysis of variance for repeated measures. Significant differences between the sham group and other groups are not shown graphically; all significance values between the sham and HI-PL-, HI 30, HI 60, or HI 90 mg/kg hIAIP-treated groups were *p* < 0.05.

**Figure 7 ijms-23-13473-f007:**
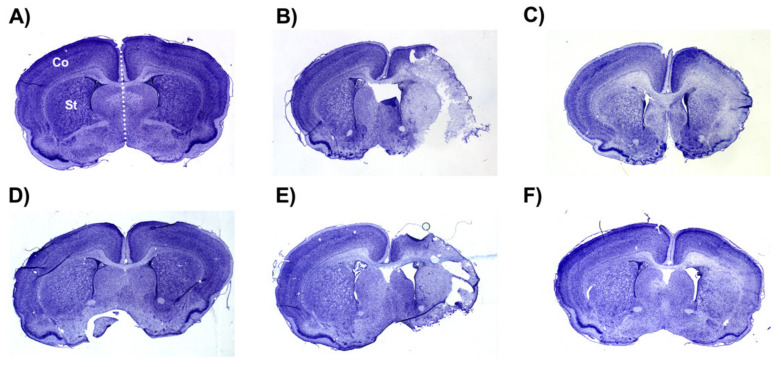
Representative images of infarct volumes in the neonatal rat brain 72 h after exposure to hypoxia–ischemia (HI). Coronal cross-sections of the brain stained with cresyl violet for female sham placebo (**A**), female HI placebo (**B**), female HI 30 mg/kg IAIP-treated (**C**), male sham placebo (**D**), male HI placebo (**E**), male HI 30 mg/kg IAIP-treated (**F**) groups. The area for cerebral cortex (Co) and striatum (St) measurements is highlighted in (**A**). Hemisphere measurements were taken to the left or right of the midline (white dotted line), as highlighted in (**A**). Figure 5 and Figure 6 show the quantification of infarct volumes for all animals of each treatment group.

**Figure 8 ijms-23-13473-f008:**
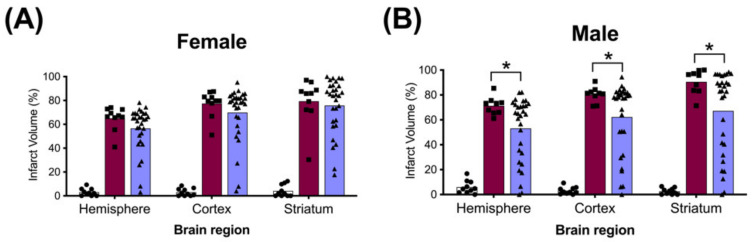
Brain infarct volumes of the sham, HI placebo, and the combined human inter-alpha inhibitor protein (hIAIP) 30, 60, 90 mg/kg treatment groups 72 h after exposure to hypoxia–ischemia (HI). The values for the infarct volumes from the hIAIP 30, 60, 90 mg/kg treatment groups were combined because significant differences between the different hIAIP dose (hIAIP 30, 60 and 90 mg/kg) regimens were not observed. Values are percent infarct volumes measured by cresyl violet staining for sham placebo- (sham, open bars, circles), HI placebo- (HI-PL, red bars, squares) and the combined hIAIP 30, 60 and 90 mg/kg (HI-hIAIP, purple bars, triangles)-treated groups. Infarct volumes for right hemisphere, right cortex and right striatum are shown for the females (**A**) and males (**B**). Each animal is presented as an individual datapoint. Statistical analysis included two-factor analysis of variance for repeated measures, F(2138) = 89.69, * *p* < 0.05. Significant differences between the sham group and other groups are not shown graphically; all significance values between the sham and HI-PL or HI-hIAIP groups were *p* < 0.05.

**Figure 9 ijms-23-13473-f009:**
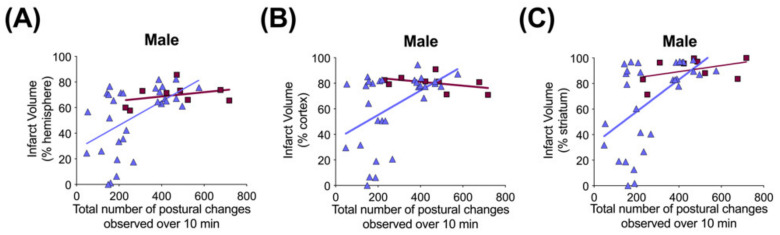
Correlation between brain infarct volume measurements and open-field postural changes in male rats 71–72 h after exposure to hypoxia–ischemia (HI) and treatment with human inter alpha-inhibitor proteins (hIAIPs). Infarct volumes are shown for the right hemisphere (**A**), right cortex (**B**) and right striatum (**C**), plotted against total number of extension and contraction postures exhibited during a 10 min open-field exploration task. Results are shown for HI placebo- (HI-PL, red squares, *n* = 9) and HI 30, 60, 90 mg/kg hIAIP (HI-hIAIP, purple triangles, *n* = 30, R = 0.46, 0.38 and 0.48, *p* < 0.05, respectively)-treated groups. Lines are linear regression.

**Figure 10 ijms-23-13473-f010:**
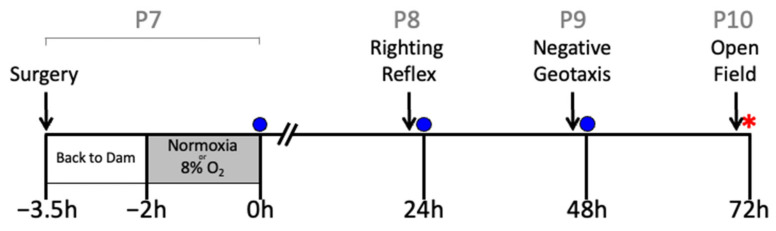
Study design. Postnatal day 7 (P7) pups were exposed to either sham surgery or right carotid artery ligation. Then, they were returned to the dam for recovery from surgery for 90 min. Thereafter, they were exposed to either HI (8% O_2,_ right carotid artery ligation groups) or normoxia (room air, sham group) for 2 h. Righting reflex (P8), negative geotaxis (P9) and small open-field (P10) analyses were performed 23, 37 and 71 h after HI, respectively. Intraperitoneal injections of human plasma-derived inter-alpha inhibitor proteins (hIAIPs, 30, 60, or 90 mg/kg) or the placebo (PBS) were injected immediately after hypoxia/normoxia (zero h) and repeated 24 and 48 h after hypoxia/normoxia. Injections (blue circles) were given after the behavioral tests had been completed. Following the conclusion of the small open-field analysis, animals were anesthetized, their blood was collected, and brains weighed and preserved for histological analysis (red star).

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
