# Peer review of "Effects of Three Different Doses of Inter-Alpha Inhibitor Proteins on Severe Hypoxia–Ischemia-Related Brain Injury in Neonatal Rats"

_ijms, 2022, doi:10.3390/ijms232113473_

Round 1
Reviewer 1 Report
Hypoxia-ischemia (HI) related brain injury in the neonates is one of the most serious complications during delivery with limited treatment modalities. In this study, Koehn and colleagues assess the neuroprotective effect of human plasma-derived inter-alpha inhibitor proteins (hIAIPs) in a neonatal rat model of severe HI brain injury, and more specifically they assess the efficacy of three different dosing levels. This study seems to be an extension of previous work from the same group which has assessed the efficacy of hIAIPs on behavioral and histological outcomes in similar models of HI encephalopathy. Consistent with their previous work, the authors confirm the therapeutic efficacy of hIAIPs with sex-specific behavioral and histological outcomes. They conclude that there are no significant differences between the three dosing groups and this is important for designing future preclinical and clinical studies.
Overall, the manuscript reads well, and the figures are precise. There are only a few minor things that should be addressed, which are listed below:
1) In the methods (line 553) the reference to IAIP extraction methods [24] seems wrong. Please correct. Authors also state that they have validated purified hIAIPs with WB, protein assay, competitive immunoassay and a trypsin inhibition assay. Given that differences in IAIP preparations between labs or batch to batch variations may explain some of the heterogeneity across experiments/studies, it would be important to offer some further characterization: Have the authors presented these data elsewhere which they can cite, otherwise can they offer as supplementary? Furthermore the authors should state whether all animals were treated with the same hIAIP batch.
2) Line 128: please offer SD of average body weight.
3) Sections 2.3.2, 2.3.3: although the authors find no effect of HI injury in negative geotaxis or some small open field measures, it would be useful for cross-studies comparisons to offer summary values for each outcome (e.g. Mean, SD).
4) Section 2.3.3: Although the authors conclude no dose-dependent effect, the sex-specific behavioral deficits (zones entered by females and posturing of males) are significantly ameliorated compared to vehicle-treated animals only for doses of 60mg/kg or higher. The Low dose group doesn’t differ from either vehicle-treated or sham-injured groups. Therefore, although not conclusive, it may be difficult to suggest that there are no differences between low and higher doses.
5) Section 2.4: Please offer representative images of infarcts in males and females indicating measured areas for hemispheres, cortex and striatum.
6) For all figures/analyses: Please report complete statistical tests e.g. f and t statistics, for each factor tested, as p-values by themselves are not useful. This could be included either in figures or as a supplementary table, if not willing to narrate in the main text.
7) Measures of serum hIAIP do not correlate with dosing; partly explaining the lack of dose response in several of the outcomes. It would be interesting to know whether there are any correlations between serum hIAIPs and histological or behavioral outcomes. Alternatively, in a secondary analysis, mice could be stratified by the achieved serum concentration (e.g. top 25% vs bottom 25%) and outcomes could be re-analysed for low and high serum hIAIPs.
8) Discussion, Lines 446-460: The variance of outcomes in the treated groups vs placebo (figures 5,6) is striking, and therefore the recognition of responders and non-responders very relevant. Are there any differences between rats that respond and those which don’t (e.g. baseline weight or serum hIAIPs)? What would be the number-needed-to-treat (NNT) for a clinically relevant outcome? And how does this compare with hypothermia treatment? Given that hypothermia is the main available clinical treatment, it would be interesting to explore whether there is therapeutic synergism between the two treatments.
9) Figures fonts and panel labels should be adjusted to be similar across figures . For consistency and clarity Fig 5,6,8 would benefit from indicating region over each panel as is done for sex in figures 1,2,3,4, 7.
Reviewer 2 Report
The present study is a novelty. The authors evaluate the different doses of Inter-alpha Inhibitor Proteins (hIAIPs) on severe hypoxia-ischemia-related brain injury in neonatal rats. They analyzed behavior by righting reflex, negative geotaxis, and small open field. It is worth mentioning that they include female neonatal models too. 30 mg/kg dose of hIAIPs was as effective as the 60 and 90 mg/kg doses. There is nothing to address. I want to congratulate the authors for their hard work.
Author Response
We appreciate the favourable remarks from reviewer 2.
Reviewer 3 Report
Koehn and colleagues studied the effect of 3 doses of human plasma-derived inter-alpha inhibitor proteins (hIAIP) in a standard hypoxic-ischemic rat neonatal model. They showed there was a slight gender differences in functional deficit to hypoxia-ischemia. The 3 doses of hIAIP used showed some response in the lower dose with no additional response in the higher doses.
There are a few major issues with this manuscript:
1) This study is very similar to the author’s original study (Chen et al., 2019, Experimental Neurology, doi: 10.1016/j.expneurol.2019.03.013.), which showed that 30 mg/kg hIAIP was an effective dose.
2) Although the current study uses two larger doses of hIAIP, the results showed that a 3 fold increase in hIAIP was not more effective, which is not a substantial finding.
3) The results provided in this manuscript was substantially less than the original study with no histological images or other in-depth cellular analysis of the brain using immunohistochemistry.
Overall, there is no major increase in scientific knowledge on the topic of hIAIP in hypoxic-ischemic neonates in comparison to what has already been published by the group for publication in IJMS with an impact factor of 6.2.
Reviewer 4 Report
Introduction: It is very long and the last paragraph in particular might be better placed within a methods section. Of note, there is no real hypothesis stated in the introduction so it makes the results difficult to evaluate. Further, there is a discussion concerning the use of cooling and one of the deficiencies of this method mentioned is its cost. However, further along, the costs are described for purifying the extract used in the study, and apparently, they are very high as well. Unless the present treatment offers a benefit over the cost of cooling, I would take it out.
Discussion: No mention of the cost of the hypothermia treatments is found here. This can likely be removed from the introduction unless addressed in the paper. The 2nd paragraph has a lot of text which can be removed as it is found in the introduction.
Methods: There is no mention of any experiments giving the sham animals the study medication and running them through the same tests. It would be interesting to see if the effects of the intervention is independent of the ischemia. For instance, when individuals without ADHD take stimulants, they have increased concentration. It would be interesting if administering the hAIP to sham animals would improve their scores over the sham animals not given hAIP. This may change the conclusions significantly. Also, the animals were given 8% O2 and a ligation. These are two different injuries and the hAIP might mitigate the effects of the global hypoxia rather than the ischemic stroke created by the ligation (or vice-versa). Please address these potential aveneus of research and how they might have affected your conclusions. Lastly, please note the stats software used.
Round 2
Reviewer 3 Report
The author provided a rebuttal to the reviewer’s comment and authors have made improvements to the manuscript. The first comment made was that the animal model was more severe. However, in both two studies (current manuscript vs Chen et al., 2019), the method was right carotid artery ligated on P7 rat then 8% O2 chamber for 2h. This was the same condition for both studies so I cannot see how this is a more severe hypoxia-ischemia model. The second comment was identifying the lowest optimal dose is important which is fair enough, but not a substantial finding. The third comment is to study the sex-specific behavioural deficits. This part of the study is worth the most and provide novelty. If this is the selling point for the paper, then the title and the writing of the manuscript should be changed to reflect this rather the 3 doses.
Fig. 7 the authors should also include the hIAIP treated group between male and female.
Reviewer 4 Report
The authors have answered all of the questions posed in the original review. The manuscript is well-written and now more clear in its objectives.
Author Response
We appreciate the favourable remarks from reviewer 4.
Round 3
Reviewer 3 Report
No further comment.